# LLM-as-a-Judge & Reward Model: What They Can and Cannot Do

## Abstract

LLM-as-a-Judge and reward models are widely used alternatives of multiple-choice questions or human annotators for large language model (LLM) evaluation. Their efficacy shines in evaluating long-form responses, serving a critical role as evaluators of leaderboards and as proxies to align LLMs via reinforcement learning. However, despite their popularity, their effectiveness in diverse contexts, such as non-English prompts, factual verification, or challenging questions, remains unexplored. In this paper, we conduct a comprehensive analysis of automated evaluators, reporting several key findings on their behavior. First, we discover that English evaluation capabilities significantly influence language-specific evaluation capabilities, often more than the language proficiency itself, enabling evaluators trained in English to easily transfer their skills to other languages. Second, we identify critical shortcomings, where LLMs fail to detect and penalize errors, such as factual inaccuracies, cultural misrepresentations, and the presence of unwanted language. Finally, we find that state-of-the-art evaluators struggle with challenging prompts in either English or Korean, underscoring their limitations in assessing or generating complex reasoning questions. We release the dataset and codes used in this research.[1]

## 1 Introduction

Automated evaluators, such as LLM-as-a-Judge and reward models (RMs), supplant human effort across a broad spectrum of large language model (LLM) research. The applications range from quality filtering of pretraining corpora (Korbak et al., 2023; Penedo et al., 2024) to replicating human preferences (Ouyang et al., 2022; Touvron et al., 2023; Lee et al., 2023) and evaluating complex model outputs (Zheng et al., 2024). As scalable al-

**Automated Evaluators Checklist**

Assess responses across different languages. ⋯⋯ ✔
Penalize factual or cultural inaccuracies. ⋯⋯⋯ ✗
Evaluate challenging prompts (i.e., GPQA) ⋯⋯⋯ ✗

Figure 1: **Summary of findings from this paper.** While LLM-as-Judge and Reward Models can be easily transferred to new languages, they struggle in penalizing cultural misrepresentations or the evaluation of challenging prompts.

ternatives to costly human annotation (Min et al., 2023; Mishra et al., 2022), they contribute to the development of user-friendly chat systems like ChatGPT (OpenAI, 2022) and Claude (Bai et al., 2022). While some research (Park et al., 2024) highlights the biases these models are prone to, there is limited understanding of their broader abilities. In this work, we examine a diverse array of automated evaluators, focusing on their failure cases.

For analysis, we create Kudge, a bilingual meta-evaluation dataset of Korean and English. The *Original* subset contains two categories: Pointwise, where a model assesses a single response on a Likert scale, and Pairwise, where the evaluator chooses the better of two responses. We also introduce the *Challenge* subset, putting emphasis on STEM questions requiring complex reasoning. Finally, we construct an ablation set with human-corrupted responses containing cultural misrepresentations or false information. This setup simulates hallucinations (Huang et al., 2023)—a common phenomenon in LLMs—to evaluate the ability of LLM judges to detect knowledge-related errors.

---

[1] https://anonymous.4open.science/r/kudge-0648

Figure 1 summarizes the key findings of this research. First, LLM-as-Judges and RMs demonstrate equal proficiency in Korean, a language they have not been trained in. Interestingly, the capability of models to evaluate Korean question and answer (QA) pairs can be predicted by their performance on REWARDBENCH (Lambert et al., 2024), an English meta-evaluation dataset. This predictive ability surpasses that of Korean language benchmarks (Son et al., 2024; 2023; Kim, 2024; Kim et al., 2022), challenging the expectation that proficient Korean speakers would naturally excel at evaluating Korean text. We conjecture that a significant portion of evaluation ability is language-agnostic, which explains this positive correlation. Secondly, however, we identify weaknesses in the transferability of proprietary and fine-tuned LLMs. These models fail to detect and penalize instances containing false information or cultural misrepresentations. This limitation indicates that they may not serve effectively as factual verifiers in unfamiliar languages or cultural contexts. Finally, automated evaluators struggle to assess QA pairs that include questions requiring complex reasoning in either language, underscoring the need for better judge models.

The main contributions of this work are as follows:

1. We empirically analyze the circumstances under which automated evaluators are effective in new languages and when not.

2. We identify the shortcomings of state-of-the-art (SOTA) judge models in evaluating challenging prompts, even in English, highlighting the need for improved evaluators.

3. We release KUDGE, a bilingual meta-evaluation dataset—the first of its kind in Korean. It also includes a challenge subset designed to enhance the evaluation of English models.

## 2 RELATED WORKS

Traditional metrics such as BLEU (Papineni et al., 2002) and ROUGE (Lin, 2004), which measure lexical overlap between texts, have long been the standard for evaluating generated text. However, as LLMs advance, they are now capable of producing semantically equivalent yet syntactically varied expressions that these metrics cannot accurately assess (Chung et al., 2023). Although human evaluation may be ideal, it is often impractical due to resource constraints (Li et al., 2023). Consequently, model-based evaluation (i.e., LLM-as-a-Judges and RMs) has emerged as an alternative, employing LLMs in a peer-review-like setup.

**Automated Evaluators**    Initially, the LLM-as-a-Judge approach involved prompting LLMs to evaluate outputs (Zheng et al., 2024; Liu et al., 2023; Verga et al., 2024). However, recent efforts have shifted towards specifically training LLMs to enhance their evaluative accuracy (Kim et al., 2024c; Vu et al., 2024; Park et al., 2024; Wang et al., 2024e). LLM-as-a-Judge models offer high flexibility, employing customizable rubrics (Ye et al., 2023) and scoring ranges (Dong et al., 2024). Typically, these models are prompted to provide feedback (Wang et al., 2024e;d) before their evaluations, which boosts performance and enhances interpretability for users. However, relying on generation for every evaluation step may be resource-intensive. Instead, reward models (RMs) attach classification heads to LLMs to directly output continuous scores (Munos et al., 2023; Wang et al., 2024c; Swamy et al., 2024). These models may be easily integrated into training systems, serving as proxies for human preferences and facilitating the training of aligned LLMs (Lee et al., 2023; Dong et al., 2023; Aksitov et al., 2023). Recent efforts also include merging the two approaches by training RMs with a next-token prediction objective (Zhang et al., 2024).

**Meta-Evaluation**    As automated evaluators gain traction, meta-evaluation tools have been developed to assess their reliability (Zeng et al., 2023; Lambert et al., 2024). These benchmarks evaluate how closely LLMs mirror human judgments and determine their efficacy as reliable proxies for human annotators. Such tools are crucial in guaranteeing the significance of LLM-based evaluations. However, despite their widespread use in multilingual settings (Aryabumi et al., 2024; Chiang et al., 2024), the full capabilities of these evaluators remain largely unexamined. This work presents a comprehensive analysis of automated evaluators across diverse settings, including non-English contexts, factual verification, and complex reasoning tasks. We assess their reliability, determine when they can be trusted, and identify key limitations in handling such scenarios.

Table 1: **An overview of the 31 LLMs used for response generation.** We use the instruct/chat version of each model if available.

| Type | Models |
|---|---|
| Proprietary | GPT-4 (Turbo-2024-04-09), HyperCLOVA X |
| Multilingual Chat | Command-R (35B, 104B), Llama-3 (8B, 70B), Gemma-1.1 (2B, 9B), Qwen-1.5 (4B, 7B, A2.7B, 14B, 32B, 72B), Yi (6B, 34B), AYA-101, ORION, Mixtral (8x7B, 8x22B) |
| English Chat | DBRX-Instruct, Falcon (7B, 40B), Mistral (7B), SOLAR (10.7B) |
| Korean Transfer | EEVE (2.8B, 10.8B), KULLM (10.7B), KORani (13B) |
| Korean Chat | 42dot-LLM (1.3B) |

# 3 KUDGE: A BILINGUAL BENCHMARK FOR AUTOMATED JUDGES

In this section, we introduce KUDGE, a dataset designed to assess the performance of automated judges. The **original** subset primarily focuses on Korean to complement existing English datasets of similar difficulty (Lambert et al., 2024; Park et al., 2024). The **challenge** subset is created in both English and Korean. Sections 3.1 to 3.3 detail the creation of the original subset, while Section 3.4 describes the development of the challenge subset. The dataset is made publicly available [2].

## 3.1 DATASET CREATION

Existing Korean benchmarks for long-form question answering (Research, 2024; Park, 2024) primarily translate or replicate MT-Bench (Zheng et al., 2024), omitting aspects of Korean culture. To address this issue, we handcraft 90 unique instructions. We classify Korean knowledge into nine distinct categories and map seven unique reasoning skills to ensure comprehensive coverage of topics and reasoning abilities, distributing the instructions evenly across these categories. We also incorporate personalized evaluation rubrics and gold-standard responses for each question. The concise size of the instruction set aims to keep evaluation and annotation feasible within budget constraints. Following this, to collect model responses of a wide variety, we generate answers for each instruction using 31 LLMs, ranging in size from 1.3 billion to over 100 billion parameters. For an overview of all models utilized in generating responses, see Table 1. Finally, fifteen human annotators, including five authors and ten hired experts compensated at ten US dollars per hour, evaluate the responses. Annotators were provided the above-mentioned reference answers and scoring rubrics to rate model generations on a Likert scale from 1 to 5. To prevent negligence, they were required to justify their scores. For quality control, each instance undergoes evaluation by two different annotators. For additional details on the creation process, see Appendix A.

## 3.2 QUALITY ANALYSIS

To assess the quality of the collected annotations, we analyze the evaluation times and agreement rates between authors and hired annotators. The average evaluation time is 146 seconds for authors and 150 seconds for hired annotators, showing no significant difference. In 83.85% of cases, annotations by the two annotators are identical or within a 1-point margin, likely due to the subjective nature of the task. However, in 8.36% of cases, disagreements exceed a 2-point margin, indicating significant discrepancies. Such discrepancies often stem from annotators assigning identical scores to multiple responses with very brief evaluation times or from unusually long annotation times exceeding 1000 seconds, which may indicate negligence or loss of concentration. In our experiments, we average scores for instances where the discrepancy between annotators is 1 point or less. Furthermore, we exclude instances with larger margins or those with only one annotation. The missing annotations are likely due to platform failures or human errors in curation.

---

[2]Removed for review.

### 3.3 POINTWISE & PAIRWISE SUBSETS

LLM-as-a-Judge applications typically include two evaluation methods: pointwise and pairwise (Vu et al., 2024; Kim et al., 2024c). Pointwise evaluation involves assigning integer scores to individual responses, whereas pairwise evaluation contrasts two responses to determine a preference. Although initially intended for pointwise assessment with a Likert scale, we have adapted KUDGE for pairwise evaluation by pairing responses from distinct models into pairs. This approach involves selecting a "chosen" response with a score exceeding three and a "rejected" response scoring two or less. Accordingly, the original subset of KUDGE features a pointwise subset comprising 2506 instances and a pairwise subset totaling 818 instances.

### 3.4 CHALLENGE SUBSET

The prompts in KUDGE are focused primarily on Korean culture, excluding STEM topics. To address this limitation, we introduce the KUDGE CHALLENGE subset, a bilingual evaluation set in English and Korean with two levels of difficulty. The "easy" level features questions from the MMLU dataset (Hendrycks et al., 2020), specifically from four categories identified by Gema et al. (2024) to have minimal issues: college physics, college mathematics, high school chemistry, and high school geography. For each question, Exaone-3-7.8B-Instruct generates 32 Chain-of-Thought (CoT) reasonings in Korean, from which we select one correct and one incorrect response. These are then translated into English. The "hard" level incorporates questions from the

Table 2: **Overview of the KUDGE dataset.** The figures in braces indicate the number of questions in Korean and English, respectively. The slight numerical differences between the two languages within the Challenge subset reflect adjustments made during translation quality checks.

| Subset | Category | # N |
|---|---|---|
| Original | Pointwise | 2506 |
| | Pairwise | 818 |
| Challenge | Pairwise-Easy | {266, 282} |
| | Pairwise-Hard | {99, 99} |

GPQA (Rein et al., 2023) dataset, following the same process but using GPT-4o. We exclude questions if neither a correct nor incorrect response emerges from the 32 CoTs, and the authors manually review translations for accuracy. We present evaluation results for this subset in Section 8. See Table 2 for an overview of the KUDGE dataset.

## 4 EXPERIMENTAL SETUP

### 4.1 PROMPT CONFIGURATION

For the evaluation of LLM-as-a-Judges, we employ a generative setting in which models are prompted to first generate an analysis and append their final decisions in a standardized format. In pointwise evaluation, models are instructed to generate the "[RESULT]" token followed by an integer from 1 to 5. In pairwise evaluation, models select the superior response as either "[[A]]" or "[[B]]", which is subsequently parsed. If a model fails to generate the correct format, we attempt up to three retries. For further details on generation configuration and prompts, refer to Appendix B. For RMs, we adopt the codebase provided from Lambert et al. (2024).

### 4.2 EVALUATED MODELS

In Table 3, we assess 20 LLMs, including three proprietary models: GPT-4o (OpenAI, 2024), Claude-3.5-Sonnet (Anthropic, 2024), and HyperCLOVA X (Yoo et al., 2024); along with five open-source model families: Llama-3.1 (Meta, 2024), Qwen 1.5/2 (Yang et al., 2024), Mistral (Mistral AI, 2024; Mistral, 2024), and Command-R (Cohere, 2024). We also include EXAONE-3.0-7.8B-Instruct (Research et al., 2024), a model pretrained on 8 trillion tokens of bilingual English and Korean text. Due to hardware constraints, a quantized version of Llama-3.1-405B is used. Additionally, in Section 6.2 we also explore fine-tuned LLM-as-a-Judge models like Prometheus2 (Kim et al., 2024c) and RMs such as FsfairX (Xiong et al., 2024), OffsetBias (Park et al., 2024), and Skywork-RM (Liu & Zeng, 2024).

Table 3: **Evaluation results for 20 LLMs on KUDGE ORIGINAL.** For the pointwise category, off-by-0.5 accuracy is considered to account for mismatches caused by averaging human annotation scores. Accordingly, random guessing accuracies are set at 30.86% for pointwise and 50% at pairwise. The highest-scoring model across all categories is highlighted in **bold**, while the top model in each category is underlined. Missing evaluation results will be updated shortly.

| Models | KUDGE Original | | | | Additional Stats. | |
|---|---|---|---|---|---|---|
| | Point (Acc) | Point (Pear.) | Pair (Acc) | Average | KMMLU | RB. |
| *proprietary language models* | | | | | | |
| GPT-4o | **61.26** | **0.62** | 87.76 | **74.51** | 64.28 | 87.28 |
| Claude-3.5-Sonnet | 56.46 | 0.59 | **89.11** | 72.79 | - | 85.17 |
| HyperCLOVA X | 51.84 | 0.55 | 85.10 | 68.47 | 53.40 | - |
| *openly available language models* | | | | | | |
| Llama-3.1-405B-Instruct (FP8) | 58.88 | 0.57 | 87.42 | 73.15 | **65.07** | **90.67** |
| Llama-3.1-70B-Instruct | 51.59 | 0.44 | 83.40 | 67.49 | 51.94 | 86.13 |
| Llama-3.1-8B-Instruct | 29.96 | 0.24 | 82.05 | 56.01 | 41.56 | 72.94 |
| Mistral-Large-Instruct | 58.44 | 0.60 | 87.93 | 73.19 | 61.17 | 88.08 |
| Mixtral-8x22B-Instruct | 49.25 | 0.46 | 84.67 | 66.96 | 47.84 | 79.84 |
| Mixtral-8x7B-Instruct | 24.48 | 0.25 | 81.51 | 53.00 | 40.61 | 74.02 |
| Mistral-Nemo-Instruct | 41.36 | 0.27 | 79.63 | 60.50 | 43.46 | 71.08 |
| Command-R-Plus | 28.34 | 0.19 | 71.98 | 50.16 | 47.84 | 73.18 |
| Command-R-v01 | 22.48 | 0.04 | 71.18 | 46.83 | 39.83 | 66.67 |
| Qwen2-72B-Instruct | 50.58 | 0.62 | 86.06 | 68.32 | 63.66 | 84.51 |
| Qwen2-7B-Instruct | 24.03 | 0.20 | 73.05 | 48.54 | 46.12 | 68.10 |
| Qwen2-1.5B-Instruct | 13.65 | 0.11 | 47.86 | 30.75 | 26.59 | 52.17 |
| Qwen1.5-72B-Chat | 15.30 | 0.32 | 76.07 | 45.68 | 51.31 | 76.60 |
| Qwen1.5-32B-Chat | 30.48 | 0.39 | 76.28 | 53.38 | 46.65 | 76.96 |
| Qwen1.5-14B-Chat | 23.54 | 0.24 | 73.75 | 48.64 | 43.16 | 68.05 |
| Qwen1.5-MoE-A2.7B-Chat | 17.69 | 0.08 | 59.02 | 38.36 | 36.75 | 51.62 |
| EXAONE-3-7.8B-Instruct | 34.58 | 0.39 | 81.86 | 58.22 | 44.94 | 67.86 |

## 5 EVALUATION RESULTS

The performance of 20 LLMs on KUDGE ORIGINAL is summarized in Table 3. Proprietary language models show the best results, with large and recent open LLMs demonstrating comparable performance. Specifically, GPT-4o (74.51) leads in overall performance, with Claude-3.5-Sonnet (72.79) trailing slightly in overall average score. Among open-weight models, Llama-3.1-405B-Instruct (73.15) and Mistral-Large-Instruct (73.19) stand out, particularly in pairwise accuracy, where they compete closely with GPT-4o. However, the Pearson correlation for the pointwise subset remains below 0.6 for most models, indicating a moderate correlation at best and suggesting significant room for improvement in this area.

Interestingly, several large models, despite their size, underperform compared to their peers. For instance, Qwen1.5-72B-Chat posts KMMLU and REWARDBENCH scores similar to Mixtral-8x22B-Instruct (51.31 vs. 47.84 in KMMLU and 76.60 vs. 79.84 in RB.), but lags by 21.28% in KUDGE (45.68 vs. 66.96). Similarly, Command-R-Plus matches Mixtral-8x22B-Instruct in other benchmarks, but falls 16.8% behind in KUDGE (50.16 vs. 66.96). This suggests that while scale remains important, as evidenced within the same family of models like the Llama-3.1 series, where larger models outperform smaller ones, it appears that newer models consistently outperform older models of similar size. This may be attributed to improvements in training data quality, larger datasets, and higher training budgets.

Figure 2 illustrates the average margin between human annotators and LLM-as-a-Judges for each score. We notice a performance gradient across score ranges. Smaller models typically struggle in the lower score range (1-2.5), demonstrating difficulty in accurately penalizing lower-quality inputs. In contrast, GPT-4o or the Llama-3.1 models tend to be conservative in awarding higher scores, as

shown by lighter shades in the higher score spectrum. Furthermore, the variance in margins per score is lowest for `GPT-4o` (0.06), followed by `Claude-3.5-Sonnet` (0.08), `Llama-3.1-405B` (0.07), and `Mistral-Large-Instruct` (0.08), correlating with performance, implying that better models tend to exhibit lower variance and consistent performance across the score range.

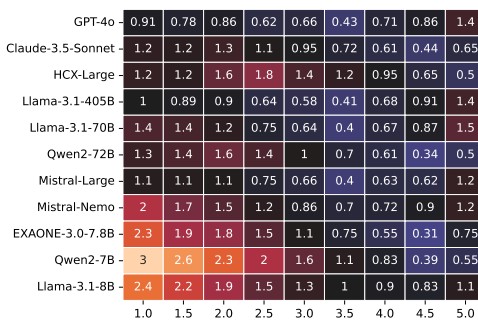

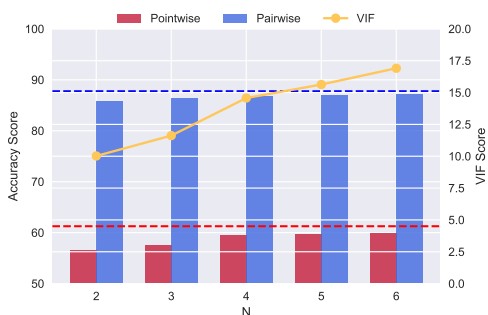

Figure 2: **Average Delta Between Human and LLM Judges**. Lighter shades represent larger deltas. The X-axis shows scores annotated by humans, while the text in the heatmap cells indicates the average margin with the scores given by models.

Figure 3: **Accuracy from Ensembling $N$ LLMs**. The table shows the average accuracy of $N$ randomly sampled LLMs across five trials. GPT-4o achieves scores of 61.26 and 87.76 on the pointwise and pairwise subsets.

## 5.1 EVALUATION WITH MAJORITY VOTING

Verga et al. (2024) suggests that aggregating judgments from multiple LLMs could enhance correlation with human evaluations. We test this hypothesis using KUDGE. In Figure 3, we sample $N$ models from proprietary and open-source LLMs with over 70B parameters, excluding the top-performing GPT-4o, and average their accuracy across five trials. Similar to prior findings, we observe aggregating models yield improvement; however, the margins are minimal and still underperform compared to `GPT-4o` alone. We find that the underperformance of aggregated models compared to `GPT-4o` alone could be attributed to multicollinearity among the models, as indicated in Figure 3. We observe that the Variance Inflation Factor (VIF) increases from 10.02 to 16.91 as more models are included in the ensemble, suggesting that these models contribute similar, rather than diverse, perspectives. This lack of diversity means the ensemble is limited in its ability to rectify incorrect assessments, undermining the effectiveness of model aggregation.

## 6 ARE JUDGING CAPABILITIES TRANSFERRED TO NEW LANGUAGES?

In Table 3, we observe that models display comparable performance despite not being explicitly trained in Korean, prompting the question: *Are judging capabilities transferable to new languages?* In this section, we investigate this issue by identifying statistically significant features that predict KUDGE scores (Section 6.1), observe whether evaluation capabilities learned in English transfer to Korean (Section 6.2) and conducting a qualitative analysis of the conditions in which transfer fails (Section 6.3).

Table 4: **Regression results for model performance on KUDGE.** $X_{(K|S)}$, and $X_{(RB|S)}$ denote Korean capabilities, and RewardBench scores, all adjusted for model size. Significance levels: ** $p < 0.05$, *** $p < 0.01$.

| Subset | $\beta_{X_{K|S}}$ | $\beta_{X_{RB|S}}$ | $R^2$ | F-Stats |
|---|---|---|---|---|
| Pointwise | 0.05 | 1.01** | 0.48 | 2.30 |
| Pairwise | 0.02 | 0.26** | 0.52 | 2.72 |

## 6.1 CORRELATION WITH DIFFERENT FEATURES

We examine the correlation of evaluation results on KUDGE with performance on KMMLU, and REWARDBENCH, as shown in Figure 4. Regression against REWARDBENCH scores yields a higher

$R^2$ value compared to KMMLU scores, suggesting that models better at English evaluations tend to perform well in Korean contexts, despite potential shortcomings in Korean-specific capabilities.

To further validate that Korean-specific abilities are less influential on Korean evaluation capabilities, we add the following benchmarks to our analysis: GSM8K-Ko, HAE-RAE Bench, and HellaSwag-KoBEST. Given the correlation in these benchmarks (Ilić, 2023; Burnell et al., 2023), we apply PCA to distill the first principal component, capturing the low-dimensional Korean capability of each model. To control for the influence of model size, which is often correlated with better scores across benchmarks, we orthogonalize both the principal component (representing Korean capabilities), and the REWARDBENCH scores with respect to model size. While controlling for computing budget

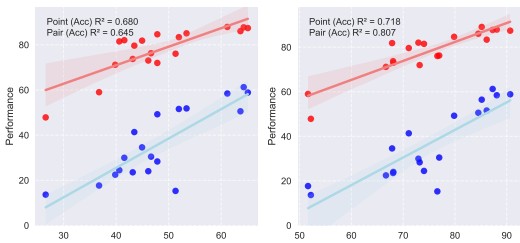

Figure 4: **Regression analysis of performance KUDGE against different benchmarks.** $R^2$ values are annotated. Pointwise scores are in blue, Pairwise score are noted in red.

would be ideal, we opt to use model size as a proxy since some open models do not disclose their training tokens, complicating the precise calculation of FLOPs used during training. The orthogonalization is denoted as $X_{K|S}$ for Korean capabilities and $X_{RB|S}$ for REWARDBENCH, where:

$$X_{K|S} = X_K - \beta_S X_S$$

and similarly for $X_{RB|S}$, where $X_K$ and $X_{RB}$ are the original scores, $X_S$ is model size, and $\beta_S$ is the coefficient from regressing the feature on size. By removing the size effect, $X_{K|S}$ and $X_{RB|S}$ represent the residual Korean and reward capabilities independent of model size. Table 4 presents the regression results. The size-adjusted language features $X_{K|S}$ show poor statistical significance, implying that Korean language capabilities have a limited impact on judging capabilities once adjusted for size. Conversely, $X_{RB|S}$ exhibits relatively higher coefficients and serves as a stronger predictor of performance in KUDGE. This observation aligns with discussions in Section 5 on why targeted training in Korean does not necessarily lead to better scores, highlighting that longer training and enhanced cognitive capabilities might be more crucial.

## 6.2 EVALUATION WITH FINE-TUNED LLMs

Recently, there have been increasing efforts to develop dedicated LLMs for evaluation, either by fine-tuning them with instruction-like data to induce judgment capabilities (Kim et al., 2024c) or by integrating a classification head and adopting a Bradley–Terry model (Wang et al., 2024b;a; Liu et al., 2024). In this section, we explore whether LLMs fine-tuned for English meta-evaluation are directly applicable as judges for Korean. In Table 5, we compare `Prometheus2` with its base model `Mistral` (Mistral, 2024). Notably, `Prometheus2` demonstrates improvements of 20.17% and 23.68% in accuracy

Table 5: **Performance comparison between Prometheus2 and its base model.** The failure column denotes how often the models fail to generate in the desired format Higher performance metrics are highlighted in **bold**.

| Model | Accuracy | Pearson | Failure |
|---|---|---|---|
| Mistral-7B | 20.29 | 0.26 | 64.33 |
| Prometheus2-7b | **46.46** | **0.43** | **7.7** |
| Mixtral-8x7B | 22.77 | 0.27 | **19.6** |
| Prometheus2-8x7b | **46.45** | **0.41** | 20.9 |

across different model sizes, suggesting that although primarily tuned in English, its meta-evaluation capabilities may effectively extend to other languages. Meanwhile, we also observe a language bias where `Prometheus` favors responses containing or written in English. In instances where `Prometheus` errs while `Mistral` is correct, the average count of English characters is 765.6, compared to 673 when the roles are reversed, indicating a preference for longer English strings. This suggests that the English-focused training of `Prometheus` introduces a subtle bias. We further investigate related biases in Section 6.3

In Table 6, we report the performance of four reward models on the pairwise subset of KUDGE ORIGINAL and compare it to their performance on REWARDBENCH. Surprisingly, reward models trained only with English data prove equally effective in Korean meta-evaluation. Notably, the top-

performing model, `FsfairX-RM-8B` (90.22), surpasses `Claude-3.5-Sonnet` (89.11), the best model evaluated in a generative setting. This suggests that training conducted in English can be effectively transferred to pairwise evaluation without additional adaptation. This observation aligns with the findings of Wu et al. (2024), which demonstrate that English RMs are also effective for cross-lingual alignment.

Interestingly, the performance rankings on KUDGE and REWARDBENCH are inversely related, with models excelling in one dataset tending to underperform in the other. We hypothesize that the performance disparities may be attributed to the size and diversity of the training datasets. For example, `FsfairX-RM-8B` utilizes the smallest dataset, while `OffsetBias-RM-8B` is a merge of a reward model trained on the OffsetBias dataset and `FsfairX-RM-8B`. Similarly, the Skywork Reward series uses a larger dataset, including the datasets from Park et al. (2024). We suspect that such data

Table 6: **Evaluation result of four RMs on KUDGE and REWARDBENCH.** The highest performance metrics are highlighted in **bold**, while the second-highest are underlined.

| Models | KUDGE | RewardBench |
|---|---|---|
| FsfairX-RM-8B | **90.22** | 84.4 |
| OffsetBias-RM-8B | 89.11 | 89.4 |
| Skywork-Reward-8B | 88.14 | 92.5 |
| Skywork-Reward-27B | 81.05 | **93.8** |

curation might lead models to overfit on English evaluations or REWARDBENCH specifically, potentially limiting their effectiveness in broader evaluation contexts. However, it should be noted that the analysis of the four models is insufficient to generalize, and future work is required to study this behavior further.

## 6.3 QUALITATIVE ANALYSIS ON BIASES

LLM-as-a-Judges are known to be susceptible to various biases (Park et al., 2024; Chen et al., 2024). To investigate such bias in KUDGE, we conduct a qualitative analysis on 15% of the pointwise subset. We identify two main types of biases: (1) **Unwanted Character**, where we find 149 responses containing non-Korean characters or sentences, and (2) **Incomplete Answers**, noted in 105 cases where responses are perceived as incomplete by humans due to refusal or low-confidence baseless claims. See Figure 9 for examples of the selected samples.

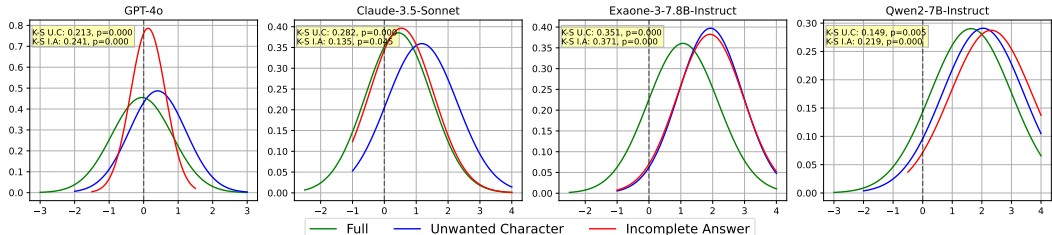

Figure 5: **Results for the Two-Sample Kolmogorov-Smirnov Test.** "U.C" denotes unwanted characters, and "I.A" stands for incomplete answers. Significance levels are indicated as follows: ** for $p < 0.05$, *** for $p < 0.01$. Analysis for remaining models are presented in Figure 15.

To determine if models struggle more when evaluating instances with such bias, we analyze their average errors, defined as the margin between model scores and human scores, across the two above-mentioned instances. If the error margins in these instances align with the model's typical error rates, it suggests that models do not specifically suffer in these scenarios. Conversely, differing error patterns indicate that models struggle with these error-prone instances.

In Figure 5, we present the cumulative distribution functions and statistics from a two-sample Kolmogorov-Smirnov (KS) test for four models. The distribution for `GPT-4o` approximates a normal distribution, indicating a balanced scoring pattern, whereas other models like `Claude-3.5-Sonnet`, `Exaone-3.7-8B`, and `Qwen2-7B-Instruct` display right-skewed patterns, suggesting a tendency to award higher scores compared to human evaluations. Specifically, the KS test reveals statistically significant differences in scoring patterns between the full evaluation set and subsets with errors such as unwanted characters or incomplete answers. These discrepancies are particularly pronounced in `EXAONE-3.0-7.8B-Instruct`, which shows the most significant distribution shift, indicating a greater sensitivity to these error types.

# 7 CAN LLMS JUDGE RESPONSES WITH FALSE INFORMATION?

While models trained in English may transfer particular learned preferences, their robustness against factual inaccuracies—easily spotted by native Korean speakers—remains uncertain. To assess this, we initially sampled 106 responses, each scoring above four from human evaluators. We exclude 18 instructions unsuitable for factual corruption, such as those requiring the creation of imaginary stories. A human annotator then corrupts the remaining responses to include false information. The annotator was required to highlight the corrupted spans and add explanations on the changes made. The corruption is performed at one of three levels: "word," involving subtle perturbations to individual words; "sentence," entailing alterations to entire sentences; and "paragraph," where broader changes are made by modifying arguments or altering comparisons. These modified responses undergo review by three human reviewers, who verify the truthfulness of each corrupted segment with a simple 'yes' or 'no'. We discarded 34 cases where the reviewers could not identify errors. Ultimately, we retain 54 instances unanimously identified as incorrect by all reviewers. Examples of perturbations are available in Appendix D.

**Feedback Analysis** As it is unclear how much we should deduct from the original score according to each corruption, instead of comparing scores, we input the corrupted responses for evaluation to two models `GPT-4o` and `Claude-3.5-Sonnet` and conduct a qualitative analysis on the generated feedback to count the number of cases where the two models succeed in identifying the errors. In Table 7, we observe that top-performing proprietary models struggle significantly in detecting factual errors. Both models perform best on *paragraph*-type errors, with `Claude-3.5-Sonnet` identifying

Table 7: **Results from a manual review on the generated feedback.** Evaluation is done in a pointwise setting using corrupted responses containing false information.

| Type | Count | GPT-4o | Claude-3.5-Sonnet |
|------|-------|--------|-------------------|
| Word | 34 | 1 (2.94%) | 0 (0%) |
| Sentence | 13 | 4 (30.8%) | 3 (21.4%) |
| Paragraph | 7 | 4 (57.1%) | 6 (85.7%) |
| Total | 54 | 9 (16.7%) | 9 (16.7%) |

nearly all, likely because these alterations significantly change the overall context, making them easily visible. However, as the errors become subtler, both models face difficulties in detection, highlighting a limitation of LLM-as-a-Judges. While they may effectively assess the logic or coherence of responses, they are less suitable for identifying the truthfulness or hallucinations in LLM outputs.

**Pairwise** In the pairwise subset, we pair corrupted responses with their original versions and present them to LLMs to assess their ability to distinguish between the two. Surprisingly, in contrast to the full subset where reward models outperformed LLM-as-a-Judges (Section 6.2), we observe an opposite trend. Reward models perform poorly, scoring below the random baseline of 50. Moreover, we manually review the feedback from the models and quantify how often the generative judges identify errors. `GPT-4o` and `Claude-3.5-Sonnet` detected the correct factual errors 26 and 32

Table 8: **Evaluations results of 4 automated evaluators using a pairwise setting.** Evaluation is done using corrupted responses containing false information.

| Type | Model | Accuracy |
|------|-------|----------|
| LLM-as-a-Judge | Claude-3.5-Sonnet | 68.52 |
| | GPT-4o | 66.67 |
| Reward Model | OffsetBias-RM-8B | 42.59 |
| | FsfairX-RM-8B | 38.89 |

times, respectively, significantly outperforming their results from Table 7. This suggests that pairwise comparisons may enhance evaluation accuracy. However, it is important to note that encountering two nearly identical responses with subtle differences is rare in real-world scenarios, making the detection of factual errors even more challenging. Additionally, we identify 13 and 16 instances where each model provided incorrect, fabricated reasons to differentiate between the responses, indicating potential robustness issues.

Table 9: **Evaluation results for 4 LLM-as-a-Judge and 4 RMs on KUDGE CHALLENGE.** Random guessing accuracies are set at 50% for all subset. The highest-scoring model across all categories is highlighted in **bold**, while the top model in each category is underlined.

| Subset | Easy | | | Hard | | |
|---|---|---|---|---|---|---|
| Language | Ko | En | Average | Ko | En | Average |
| *openly available language models (>70B)* | | | | | | |
| Llama-3.1-405B-Instruct (FP8) | 70.92 | 83.08 | 77.00 | 53.53 | **64.64** | **59.09** |
| Llama-3.1-70B-Instruct | 68.79 | 71.80 | 70.30 | **63.63** | 48.48 | 56.06 |
| Qwen-2-72B-Instruct | 71.73 | 78.94 | 75.34 | 30.30 | 41.41 | 35.86 |
| Mistral-Large-Instruct | 74.11 | 79.32 | 76.72 | 46.46 | 53.53 | 50.00 |
| *reward models* | | | | | | |
| Skywork-Reward-27B | **80.49** | 81.58 | 81.04 | 9.09 | 10.10 | 9.60 |
| Skywork-Reward-8B | 72.34 | 72.93 | 72.64 | 12.12 | 18.18 | 15.15 |
| OffsetBias-RM-8B | 75.17 | 73.68 | 74.43 | 19.19 | 34.34 | 26.77 |
| FsfairX-RM-8B | 76.59 | 81.20 | 78.90 | 17.17 | 19.19 | 18.18 |

# 8  CAN LLMS EVALUATE CHALLENGING PROMPTS?

Intuitively, evaluating a problem necessitates solving it, as the evaluator must determine the correctness of an answer. We posit that models may find it difficult to assess challenging questions that they themselves cannot solve. In this section, we use the KUDGE CHALLENGE subset to explore how question difficulty influences judgability.

In Table 9, we observe that model performance significantly correlates with the difficulty of the prompts. All models manage reasonable success on simpler MMLU questions but encounter substantial difficulties with the more demanding GPQA subset. No model surpasses a 70% accuracy rate on these harder questions, a concerning performance given that random guessing would result in 50% accuracy. Reward models, although competitive on easier sets, underperform on tougher questions, likely due to their relatively smaller size (the largest being 27B parameters) compared to the LLMs-as-judges, all of which exceed 70B parameters. This disparity suggests that the complexity of GPQA questions particularly challenges smaller models. Overall, the findings indicate that models struggle with complex questions, revealing a significant limitation in employing state-of-the-art, open-source judges for training LLMs (OpenAI, 2024) on advanced reasoning tasks.

# 9  CONCLUSION

In this paper, we introduce KUDGE, a bilingual meta-evaluation dataset that assesses meta-evaluation capability in both Korean and English. Our findings show that LLM-as-a-Judges and RMs perform comparably in Korean, effectively evaluating questions and answers beyond their English training. This observation challenges the assumption that native Korean proficiency is essential for high-quality text evaluation. However, we also uncover significant deficiencies in proprietary and finely-tuned models' ability to detect cultural misrepresentations or false information. This underscores a substantial gap in their effectiveness as reliable fact-checkers across varied languages and contexts. Furthermore, both model types struggle with complex reasoning tasks in our *Challenge* subset, pointing to a widespread limitation in current automated evaluators.

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

## A ADDITIONAL DETAILS OF KUDGE

### A.1 K2-EVAL

Given the absence of long-form question datasets in Korean at the time of this research, we create K2-EVAL, a curated set of 90 prompts encompassing various aspects of Korean knowledge. The concise dataset size is intended to keep evaluation and annotation manageable within budgetary limits. Despite its small size, we ensure broad coverage by categorizing Korean knowledge into nine distinct areas and identifying seven unique reasoning skills, with each task in the dataset pairing one knowledge category with one reasoning skill. See Table 10 for an overview of the knowledge and reasoning types. The gold answer for each instruction is crafted through a three-step process. Initially, we generate answers for each prompt using GPT-4, enhanced with a browsing-augmented chain of thought (CoT) technique (Yao et al., 2022). Subsequently, two authors independently review and amend any problematic responses. Finally, one author consolidates these revisions to finalize the gold standard answers. We also establish specific scoring rubrics and evaluation metrics for each task, linked to the paired knowledge and reasoning categories. These assessment tools measure the effectiveness of responses, their cultural accuracy, and the use of unique Korean linguistic elements, including honorifics. The scoring rubric and evaluation criteria are not unique to each instruction instead, they are shared within each combination of knowledge and reasoning types.

Table 10: **Summary of the knowledge and reasoning types defined for instruction creation.**

| Category | Subcategory | Description |
| --- | --- | --- |
| Knowledge | Art | Traditional and contemporary Korean art, including historical context. |
| Knowledge | Culinary | Knowledge of traditional Korean foods, recipes, and food culture. |
| Knowledge | Culture & Traditions | Understanding of diverse cultural practices in Korea. |
| Knowledge | Geography | Korean natural environments, topography, and architectural influence. |
| Knowledge | History | Recognition of historical events, and figures from ancient to modern times. |
| Knowledge | Linguistics | Comprehension of Korean linguistic characteristics and dialects. |
| Knowledge | Politics & Economy | Understanding of government systems, and economic policies. |
| Knowledge | Social Issues | Awareness of contemporary social challenges in Korean society. |
| Reasoning | Empathetic Reasoning | Ability to understand and interpret others' emotions and perspectives. |
| Reasoning | Brainstorming | Capacity for divergent thinking and generating creative solutions. |
| Reasoning | Cause & Effect Analysis | Skill in identifying and analyzing causal relationships between events. |
| Reasoning | Comparative Analysis | Ability to compare and contrast subjects to evaluate similarities and differences. |
| Reasoning | Numerical Estimation | Competence in making mathematical estimations in data-limited scenarios. |
| Reasoning | Creative Writing | Capability to generate original narratives and use various literary devices. |
| Reasoning | Proposing Solutions | Skill in suggesting feasible solutions to problems within realistic constraints. |

In Figure 6, we examine the relationship between response length and the average score annotated by humans. We observe a slight trend where higher scores correlate with longer average lengths, however not to a concerning extent.

### A.2 HUMAN ANNOTATION

A group of 15 human annotators, consisting of authors and 10 hired experts, all native Korean speakers with at least a bachelor's degree from a Korean university, were recruited for this task. The group included a diverse mix of seven females and eight males. Annotators were tasked with scoring model responses on a Likert scale from 1 to 5, using a provided scoring rubric, a reference answer representing a score of 5, and specific evaluation criteria. To ensure consistent evaluation standards,

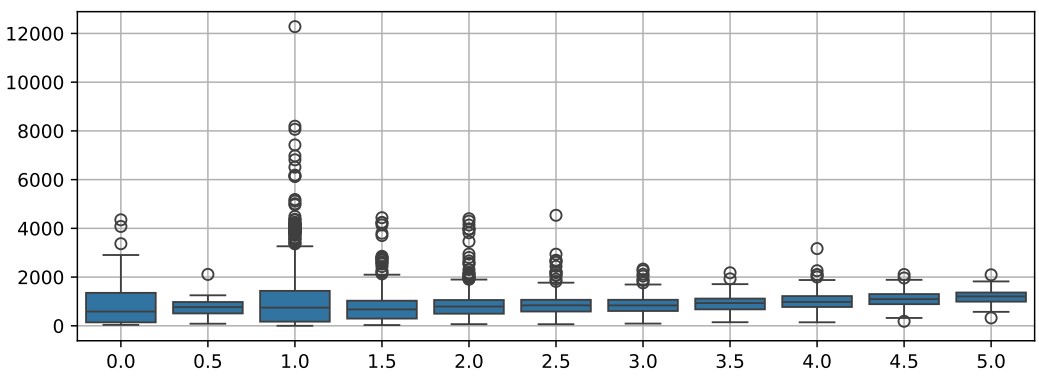

Figure 6: **An analysis between length and human scores.** The X-axis represents the average scores annotated by human reviewers, while the Y-axis shows the response lengths in number of characters.

annotators received a 30-minute training session with examples from the dataset. To minimize human error, each instance was independently annotated by two different annotators, securing two sets of evaluations per instance.

We observe that annotators completely agree only 52.2% of the time. 30.7% of the time, they assign scores with a one-point difference. Surprisingly, despite the detailed evaluation criteria and rubrics provided, 17.1% of the time annotators disagree by a margin bigger than two. In our work, we treat cases where scores differ by one point as agreements and calculate the final score by averaging these two scores. However, in instances where the score difference is two points or more, occurring 17% of the time, we consider these significant disagreements and exclude them from our analysis.

Figure 7 presents a cross-tabulation of the scores given by the two annotators, normalized column-wise. We observe that the agreement rate is highest for the lowest score and decreases progressively towards the highest score. This trend suggests that while annotators commonly agree on identifying poor responses, they often differ on recognizing high-quality responses.

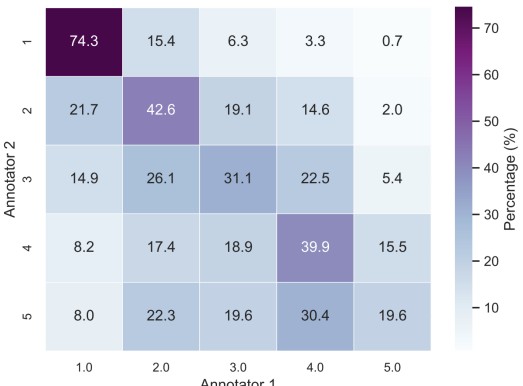

Figure 7: **Agreement rate of the two annotators.** Note that the values are normalized column-wise.

### A.3 Benchmark results

We leverage 31 LLMs, mentioned in Table 1, to generate responses for the K2-Eval dataset and subsequently hire annotators to evaluate these responses. Figure 8 presents the performance of these 31 models as evaluated by humans. `HyperCLOVA X` shows the highest performance, followed closely by `GPT-4` and `Command-R-Plus`. Interestingly, smaller models specifically fine-tuned on Korean instruction data, such as `EEVE-Korean-Instruct-10.8B` (Kim et al., 2024b) and `KULLM3` (Kim et al., 2024a), outperform larger counterparts like `Mixtral-8x22B-Instruct` and `Qwen-1.5-72B-Chat`. This underscores the significance of localized tuning, which addresses linguistic and cultural nuances to enhance performance in terms of human preference, beyond mere model size.

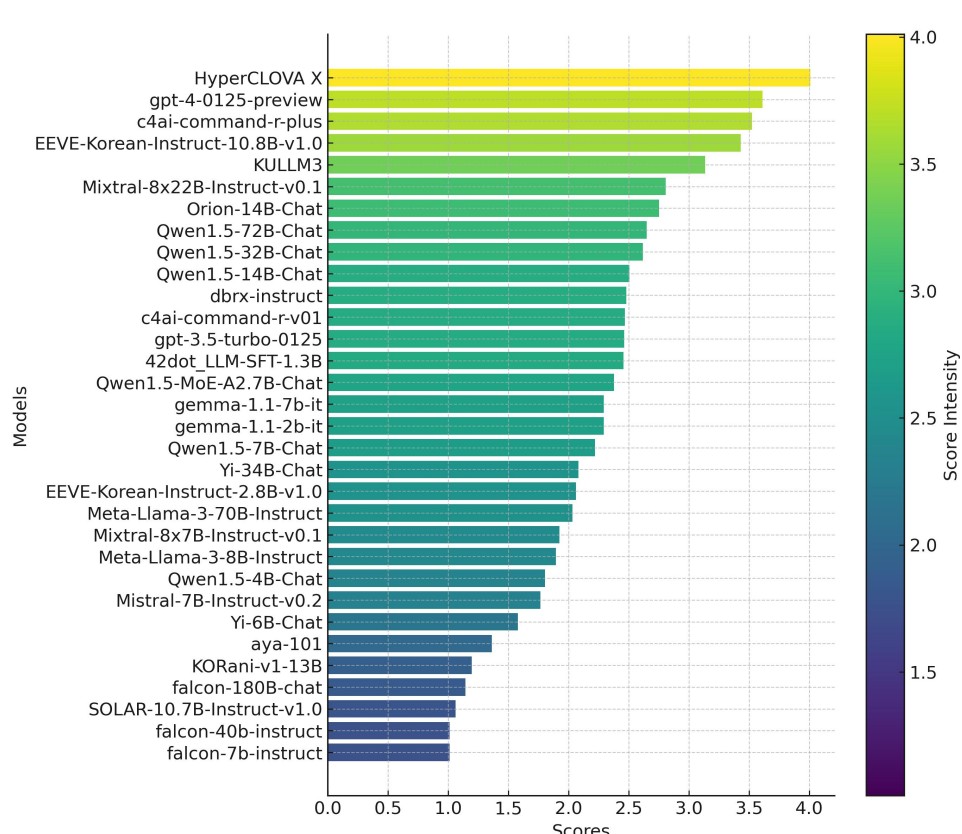

Figure 8: **K2-EVAL human evaluation result**.

## B EVALUATION PROMPTS

For evaluation, we use the following prompting formats.

**Pointwise Evaluation**

```
###Task Description:
An instruction (might include an Input inside it), a response to
    evaluate, a reference answer that gets a score of 5, and a
    score rubric representing a evaluation criteria are given.
1. Write a detailed feedback that assess the quality of the
    response strictly based on the given score rubric, not
    evaluating in general.
2. After writing a feedback, write a score that is an integer
    between 1 and 5. You should refer to the score rubric.
3. The output format should look as follows: "Feedback: (write a
    feedback for criteria) [RESULT] (an integer number between 1
    and 5)"
4. Please do not generate any other opening, closing, and
    explanations.
5. Respond in fluent Korean.

###The instruction to evaluate:
{orig_instruction}
```

```
###Response to evaluate:
{orig_response}

###Reference Answer (Score 5):
{orig_reference_answer}

###Score Rubrics:
{score_rubric}

###Feedback:
```

**Pairwise Evaluation**

```
Please act as an impartial judge and evaluate the quality of the
    responses provided by two AI assistants to the user question
    displayed below. You should choose the assistant that follows
    the user's instructions and answers the user's question better.
     Your evaluation should consider factors such as the
    helpfulness, relevance, accuracy, depth, creativity, and level
    of detail of their responses. Begin your evaluation by
    comparing the two responses and provide a short explanation.
    Avoid any position biases and ensure that the order in which
    the responses were presented does not influence your decision.
    Do not allow the length of the responses to influence your
    evaluation. Do not favor certain names of the assistants. Be as
     objective as possible. After providing your explanation,
    output your final verdict by strictly following this format:
    "[[A]]" if assistant A is better, "[[B]]" if assistant B is
    better.

### Instruction:
{instruction}

### Response A:
{response_a}

### Response B:
{response_b}

###Feedback:
```

## C  EXAMPLES FROM QUALITATIVE ANALYSIS

In Section 6.3, we identify two common error types that humans easily detect but models frequently struggle: **Unwanted Characters** and **Incomplete Answers**. Examples of these biases, as identified by human annotators, are illustrated in Figure 9.

In Section 7, authors manually conduct reviews on model feedback to assess their ability to detect factual errors. In Figures 10 and 11, we present examples of cases where the model detects an error in its feedback.

## D  FURTHER DETAILS ON FACTUAL PERTURBATIONS

In Section 7, we manually add three types of corruption to KUDGE ORIGINAL to create a factually corrupted subset. In Figures 12 to 14 provide examples of the perturbations.

**Unwanted Charecters**

저출산 probleM(Low Fertility Problem)은 Korea(South Korea)가 recent years(오랜 시간)에 서 serious challenge(심각한 Challenges)로 인정됨. Current Situation(현 Status)은 Korea's Total Fertility Rate(Korea의 총 출생 Rate)가 1.2(2019년 base)로 1.3(population replacement level)을 significantly undershoot(크게 미달)함. [...]

**Incomplete Answers**

As an AI language model, I cannot have my own opinions or beliefs. However, I can provide some sugges-
tions to help you analyze the issue of coal mining in Korea and propose policies to solve it:
1. Analyze the current situation of coal mining in Korea: You can start by understanding the current situation of coal mining in Korea [...]

Figure 9: Example of a model response with non-Korean charecters

**Feedback by GPT-4o**

Both responses attempt to analyze and compare the works 광장 by 최인훈 and 무진기행 by 김 승옥. They discuss the relationship between individuals and society, focusing on themes of internal struggle, societal pressures, and the reflection of 1960s-70s Korean society in each novel. However, there are crucial factual errors in Response B. Response B incorrectly attributes 광장 to 김승옥 and 무진기행 to 최인훈, while these works are generally known to be authored by 최인훈 and 김승옥 respectively. This error significantly affects the clarity and reliability of the analysis provided. [...]

Figure 10: Example where `GPT-4o` detects a factual error in the response within its feedback.

## E  ADDITIONAL RESULTS

In this section, we present additional experimental results. Figure 15 displays the outcomes of the Two-Sample Kolmogorov-Smirnov Test on the remaining models, continuing from Section 6.3.

**Feedback by Claude-3.5-Sonnet**

응답 A와 B를 비교해보면, 두 응답 모두 한국 사회의 높은 교육열과 과도한 사교육 문제에 대한 원인, 결과, 그리고 해결책을 제시하고 있습니다. 그러나 응답 A가 더 정확하고 현실적인 분석을 제공하고 있습니다. 응답 A는 입시 위주의 교육 체계를 한국의 교육 문제의 주요 원인으로 정확히 지적하고 있습니다. 반면 응답 B는 "기초 과학 위주의 교육 체계"와 "노벨상 수상"을 언급하고 있는데 , 이는 한국의 교육 현실과 맞지 않는 부정확한 분석입니다.[...]

**English Translation**

When comparing responses A and B, both address the causes, effects, and solutions to the high educational fervor and excessive private tutoring in Korean society. However, response A provides a more accurate and realistic analysis. Response A correctly identifies the exam-oriented education system as the main cause of educational issues in Korea. On the other hand, response B mentions an "education system focused on basic sciences" and "Nobel Prize winning," which are inaccurate analyses that do not align with the realities of education in Korea.[...]

Figure 11: Example where `Claude-3.5-Sonnet` detects a factual error in the response within its feedback. Translations are provided for broader accessibility.

✳ The Japanese Colonial Period started in 1910.

**Original Response**

일제 강점기(1910-1945) 는 한국 역사에서 매우 어두운 시기로, 일본의 식민 지배는 한국의 정치, 경제, 사회, 문화 등 여러 분야에 깊은 영향을 미쳤습니다. 특히, 한일 정치 관계는 한국어 정책과 교육에 직접적인 영향을 끼쳤으며, 그 여파는 현대 한국어에도 여전히 남아 있습니다. [...]

The Japanese Colonial Period (1910-1945) is a very dark time in Korean history. Japan's colonial rule deeply influenced various sectors in Korea such as politics, economy, society, and culture. Particularly, the political relations between Korea and Japan had a direct impact on Korean policies and education, and its effects still remain in modern Korea.

**Corrupted Response**

일제 강점기(1110-1945) 는 한국 역사에서 매우 어두운 시기로, 일본의 식민 지배는 한국의 정치, 경제, 사회, 문화 등 여러 분야에 깊은 영향을 미쳤습니다. 특히, 한일 정치 관계는 한국어 정책과 교육에 직접적인 영향을 끼쳤으며, 그 여파는 현대 한국어에도 여전히 남아 있습니다. [...]

The Japanese Colonial Period (1110-1945) is a very dark time in Korean history. Japan's colonial rule deeply influenced various sectors in Korea such as politics, economy, society, and culture. Particularly, the political relations between Korea and Japan had a direct impact on Korean policies and education, and its effects still remain in modern Korea.

Figure 12: Example of a word level corruption. Additional details on the corruption is provided in red. Translations are provided for broader accessibility.

---

✳ The explanation of the two poets "Yun Dong Ju" and "Baek Seok" has been reversed.

**Original Response**

두 시인은 일제 강점기라는 시대적 상황을 경험했지만, 그들의 시 세계는 사뭇 다르게 나타납니다. 윤동주가 시대의 아픔을 절망과 우울, 그리고 항거로써 표현했다면, 백석은 그 상황 속에서도 일상의 아름다움과 소중함을 노래했습니다. 그러나 그들의 시에는 공통적으로 잃어버린 조국에 대한 사랑과 그리움이 녹아있으며, 이는 암울한 시대에 위로와 희망을 주었습니다.

Although the two poets experienced the era of Japanese colonial rule, their poetic worlds are distinctly different. While Yun Dong-ju expressed the pain of the times through despair, depression, and resistance, Baek Seok sang about the beauty and preciousness of everyday life even in those circumstances. However, in their poetry, there is a common theme of love and longing for the lost homeland, which provided comfort and hope during the dark times.

**Corrupted Response**

두 시인은 일제 강점기라는 시대적 상황을 경험했지만, 그들의 시 세계는 사뭇 다르게 나타납니다. 윤동주가 시대의 아픔과 절망 속 희망 및 행복, 그리고 수용을 표현했다면, 백석은 일상의 아름다움의 상실과 허무감을 노래했습니다. 그러나 그들의 시에는 공통적으로 잃어버린 조국에 대한 사랑과 그리움이 녹아있으며, 이는 암울한 시대에 위로와 희망을 주었습니다.

Although the two poets experienced the era of Japanese colonial rule, their poetic worlds are markedly different. Yun Dong-ju expressed hope and happiness amidst the pain and despair of the times, and acceptance, whereas Baek Seok sang about the loss of everyday beauty and a sense of emptiness. However, their poetry commonly embodies love and longing for the lost homeland, providing comfort and hope during the bleak era.

---

Figure 13: Example of a sentence-level corruption. Additional details on the corruption are provided in red. Translations are provided for broader accessibility.

✱ Explanation of artistic styles in Korea and France has been reversed throughout the entire passage.

**Original Response**

전통적인 한국식 정원과 프랑스 바로크식 정원을 설계하는 원칙을 비교하는 것은 매우 흥미롭습니다. 대칭성, 공간 활용도, 철학적 근거와 같은 측면에 초점을 맞추면서 이러한 정원이 각 문화의 미학과 가치를 어떻게 반영하는지에 대해 논의해 보겠습니다.
대칭성은 두 정원의 가장 큰 차이점 중 하나입니다. 한국식 정원은 자연의 불균형을 반영하기 위해 대칭성을 피하는 경향이 있습니다. 반면, 프랑스 바로크식 정원은 완벽한 비율과 균형을 추구하며 대칭성이 중요합니다. 이는 동양과 서양의 미학적인 관점에서 비롯된 것입니다. 동양에서는 불완전함, 즉 자연스러움이 미덕으로 간주되는 반면, 서양에서는 균형과 정확성이 더 중요시됩니다.

Comparing the design principles of traditional Korean gardens and French Baroque gardens is very intriguing. We will discuss how these gardens reflect the aesthetics and values of each culture by focusing on aspects such as symmetry, spatial utilization, and philosophical foundations.
Symmetry is one of the biggest differences between the two types of gardens. Korean gardens tend to avoid symmetry to reflect the asymmetry of nature, whereas French Baroque gardens seek perfect proportions and balance, and symmetry is crucial. This originates from the aesthetic perspectives of the East and West. In the East, imperfection, or naturalness, is considered a virtue, while in the West, balance and precision are more valued.

**Corrupted Response**

전통적인 프랑스식 정원과 한국 바로크식 정원을 설계하는 원칙을 비교하는 것은 매우 흥미롭습니다. 대칭성, 공간 활용도, 철학적 근거와 같은 측면에 초점을 맞추면서 이러한 정원이 각 문화의 미학과 가치를 어떻게 반영하는지에 대해 논의해 보겠습니다.
대칭성은 두 정원의 가장 큰 차이점 중 하나입니다. 프랑스식 정원은 자연의 불균형을 반영하기 위해 대칭성을 피하는 경향이 있습니다. 반면, 한국 바로크식 정원은 완벽한 비율과 균형을 추구하며 대칭성이 중요합니다. 이는 동양과 서양의 미학적인 관점에서 비롯된 것입니다. 동양에서는 불완전함, 즉 자연스러움이 미덕으로 간주되는 반면, 서양에서는 균형과 정확성이 더 중요시됩니다.

Comparing the design principles of traditional French gardens and Korean Baroque gardens is very interesting. We will discuss how these gardens reflect the aesthetics and values of each culture by focusing on aspects such as symmetry, spatial utilization, and philosophical foundations.
Symmetry is one of the biggest differences between the two types of gardens. French gardens tend to avoid symmetry to reflect the asymmetry of nature, whereas Korean Baroque gardens seek perfect proportions and balance, with symmetry being crucial. This stems from the aesthetic perspectives of the East and West. In the East, imperfection, or naturalness, is considered a virtue, while in the West, balance and precision are more highly valued.

Figure 14: Example of a word-level corruption. Additional details on the corruption are provided in red. Translations are provided for broader accessibility.

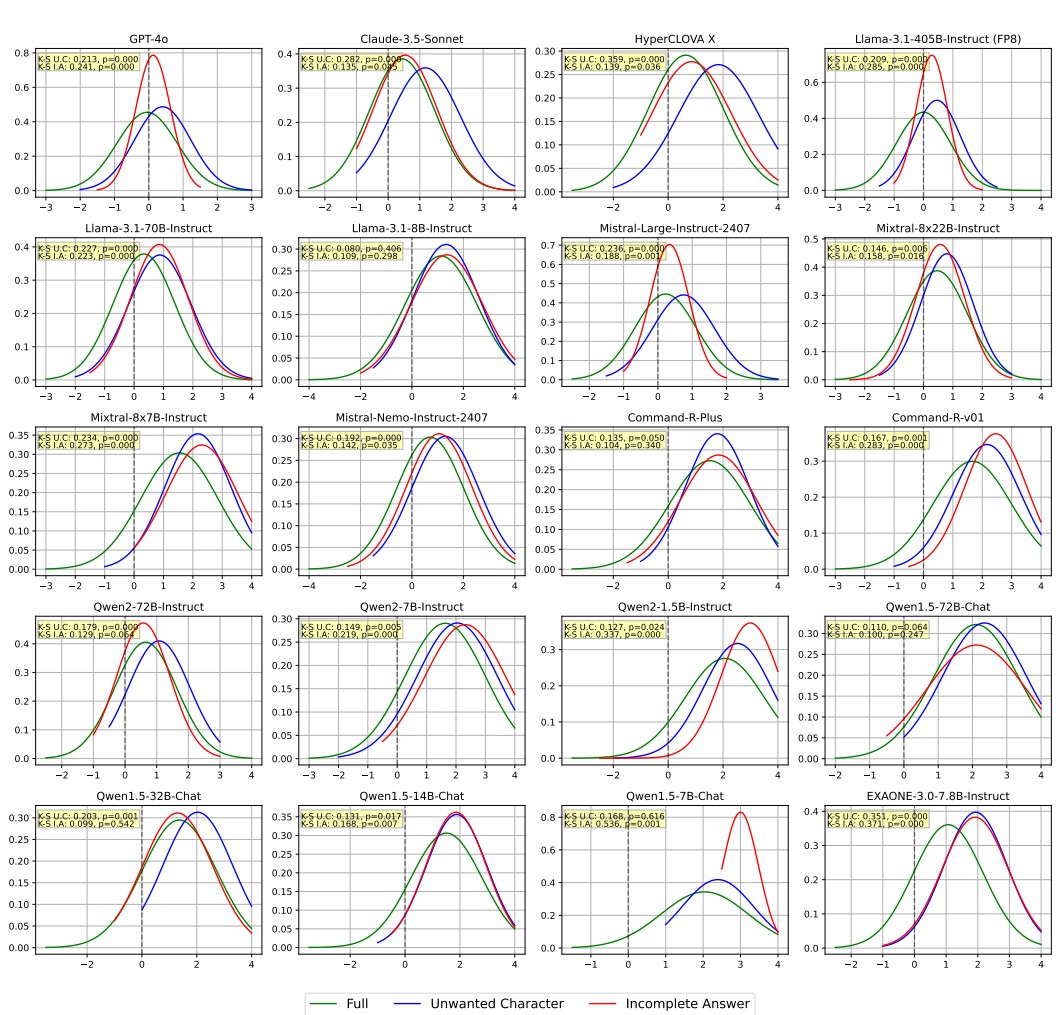

Figure 15: **Results for the Two-Sample Kolmogorov-Smirnov Test.** "U.C" denotes unwanted characters, and "I.A" stands for incomplete answers. Significance levels are indicated as follows: ** for $p < 0.05$, *** for $p < 0.01$.

