# OpenReview forum: "LLM-as-a-Judge & Reward Model: What They Can and Cannot Do"
_ICLR.cc/2025/Conference — ICLR 2025 Conference Withdrawn Submission_

### Official Review · Reviewer_NEjL · 2024-10-30

**Soundness:** 3
**Presentation:** 3
**Contribution:** 2
**Rating:** 5
**Confidence:** 2

**Summary:**

The paper evaluates LLMs as judges focusing on their performance in new languages such as Korean. The authors conduct a comprehensive analysis of these models, focusing on how well their judgment abilities transfer to new languages and contexts that they were not explicitly trained on. Key findings indicate that while these models exhibit notable cross-lingual transfer, enabling them to evaluate Korean responses reasonably well based on English training, they still face significant challenges in certain areas.

**Strengths:**

- The evaluation is very comprehensive, covering all kinds of models, either open sourced or proprietary models.

**Weaknesses:**

- The title is too broad: The title of the paper is too broad relative to the specific content covered. While the study centers on the effectiveness of LLMs in judging new languages, a more precise title would better encapsulate the paper’s focus and scope.
- The conclusion is too generalized: The conclusions drawn in the paper are overly generalized, as the authors only evaluate Korean in addition to English. To make the conclusions stronger, it would be good to include evaluation of additional languages to really conclude the circumstances under which automated evaluators are effective in new languages and when not.

**Questions:**

- The paper claimed that the pretrained models are not trained on Korean data. However, this claim is unclear due to the lack of detailed information regarding the pretraining datasets of these proprietary models.
- The reward model shows poor performance in questions related to cultural misrepresentations. Is this level of performance expected given the models are not trained to be multilingual. What are the new insights we can gain from these findings?
- The paper always claims "across languages" but only evaluates Korean

---

### Official Review · Reviewer_bTRP · 2024-10-31

**Soundness:** 2
**Presentation:** 2
**Contribution:** 2
**Rating:** 5
**Confidence:** 3

**Summary:**

This paper presents a meta-evaluation dataset designed to assess the performance of current LLM judges and reward models. The authors demonstrate that evaluation capabilities in English can be effectively transferred to other languages. Additionally, they highlight several limitations of existing LLM judges and reward models, including their inability to accurately identify factual errors and their struggles with more challenging prompts.

**Strengths:**

1. The paper is well motivated. It argues that there is a lack of understanding of current LLM judges and reward models, which are mostly tested on English benchmarks with general instructions-following prompts.

2. By compiling a meta-evaluation dataset, the paper makes a valuable contribution that is likely to influence future research.

3. The paper has revealed some very interesting findings, such as the transferability of English-focused LLM judges and reward models. It demonstrates that these models can effectively extend their evaluation capabilities to other languages, such as Korean.

**Weaknesses:**

The primary concern with this paper is its attempt to address too many questions simultaneously and some of the findings are not particularly surprising. Specifically, it seeks to explore whether LLM judges and reward models can transfer their evaluation capabilities from English to other languages, assess how well these models can identify factual errors, and evaluate their performance on challenging prompts. Due to the breadth of these inquiries, the paper lacks a deep understanding of each individual question.

- Regarding the question of transferability, the claim that LLM judges and reward models can "transfer their skills to other languages" is difficult to substantiate, as the study only tests this on Korean. Additionally, the authors do not provide a detailed examination of the training data for these models.

- Concerning the issue of factuality, the findings are not particularly surprising, given that LLMs are known to hallucinate frequently. If a model inherently holds incorrect beliefs about certain factual knowledge, it is unlikely to assess factuality accurately. Moreover, there are existing works that address factuality error detection through search-augmented evaluations, such as [1][2].

- As for the performance on challenging prompts, it is unsurprising that reward models struggle with problems in domains like biology, physics, and chemistry, as they are not specifically trained for these areas.



[1] FacTool: Factuality Detection in Generative AI -- A Tool Augmented Framework for Multi-Task and Multi-Domain Scenarios

[2] Long-form factuality in large language models

**Questions:**

Regarding language transferability, does this imply that for evaluating Korean responses, it is preferable to use English-focused LLM judges and reward models? It would be interesting to compare the performance of these top-performing English-focused models against top-performing Korean-focused LLM judges and reward models. Such a comparison could provide valuable insights. However, it appears that in the paper, only two LLMs—HyperCLOVA X and EXAONE-3—have been trained on a substantial amount of Korean text.

---

### Official Review · Reviewer_eGhH · 2024-11-02

**Soundness:** 2
**Presentation:** 2
**Contribution:** 2
**Rating:** 3
**Confidence:** 3

**Summary:**

This paper presents an analysis of automated evaluators (LLM-as-a-Judge and reward models) in assessing language model outputs. The study focuses on understanding these evaluators' capabilities and limitations across two languages (English, Korean), factual verification, and complex reasoning tasks. The authors introduce KUDGE, a bilingual meta-evaluation dataset in Korean and English, and conduct extensive experiments to analyze how these automated evaluators perform in various scenarios. The key findings reveal that while evaluation capabilities transfer well across languages, these models struggle with factual verification, cultural misrepresentations, and complex reasoning tasks.

**Strengths:**

- Cross-Lingual Analysis: The paper provides insights into how evaluation capabilities transfer across languages, showing that English evaluation proficiency is a stronger predictor of Korean evaluation performance than Korean language proficiency itself. This finding challenges conventional assumptions and has important implications for developing multilingual evaluation systems.

- Comprehensive Evaluation Framework: The authors create a well-structured/controlled evaluation dataset (KUDGE) that covers multiple dimensions: pointwise vs. pairwise evaluation, original vs. challenge subsets, and includes carefully crafted perturbations for testing factual verification.

- Extensive Evaluation of different models: The study employs a diverse set of 31 LLMs ranging from 1.3B to over 100B parameters, ensuring comprehensive coverage of model scales and architectures.

**Weaknesses:**

- **Lack details about the quality control process**: The authors discuss quality analysis on Section 3.2, but did not mention the exact inter-rater reliability (e.g., fleiss’s kappa), which will be essential to gague the quality of the dataset. Also, there lacking discussion for the process where challenge subset is constructed: “which we select one correct and one incorrect response. These are then translated into English” – is the responses being randomly selected? How is translation process completed? “the authors manually review translations for accuracy” - what is the inter-rater agreement here (e.g. does the authors almost always agree on a subset of data)?


- **Limited insights from the paper**: the authors list findings & observations, yet there's little actionable insights that would allow us to build a better system in the future, rather than "it doesn't work."
For example, about the correlation between “English evaluation proficiency and Korean evaluation performance.” (Figure 4).  This is rather unsurprising – It is the same task (i.e., judging) with a different language. Rather the correlation between judging performance and KMMLU compares the performance of different tasks, which it is expected to have lower correlation. Furthermore, these results are based on a single language pair, so it is unclear if this trend will generalize across languages (would be interesting if so). I would suggest to further analyzing the results to back up concrete claims that would be helpful for developing better systems, and bring these up in the introduction.


- **Limited language pairs & sample size**: While the study provides valuable insights about Korean-English transfer, it only focuses on this language pair. Even within Korean, with a limited sample size (90 instructions as mentioned in L135), it is unclear whether the claim that “English evaluation proficiency is a stronger predictor of Korean evaluation performance than Korean language proficiency itself” would hold across different languages. Including more languages (at least one more), especially those from different language families, would help demonstrate the generalizability of the findings.

**Questions:**

- Can you provide more insights on how the findings from this paper would impact the development of better future systems? i.e., what would be the actionable insights from this paper that can help us build better system?

- How do the findings about language transfer capabilities generalize to languages that are more distant from English than Korean? Would the same patterns hold for languages with very different grammatical structures or writing systems?

---

### Official Review · Reviewer_pwZD · 2024-11-06

**Soundness:** 3
**Presentation:** 2
**Contribution:** 2
**Rating:** 3
**Confidence:** 4

**Summary:**

The authors conduct a comprehensive analysis of a wide range of automated evaluators, with a particular emphasis on their failure cases. Additionally, they introduce a novel bilingual meta-evaluation dataset designed to assess meta-evaluation capabilities in both Korean and English. This study delves into the effectiveness of automated evaluators, with a focus on LLM-as-a-Judge and reward models. Furthermore, the authors present significant insights into the behavior of these evaluators, elucidating both their strengths and weaknesses.

**Strengths:**

1. The motivation around the LLM-as-a-Judge and reward models are very important and more related exploratory work is needed.
2. The creation of a bilingual meta-evaluation dataset (KUDGE Dataset) is a significant contribution, especially in the context of non-English languages like Korean.
3. The paper conducts a thorough analysis of automated evaluators across diverse contexts, providing valuable insights.

**Weaknesses:**

1. The evidence of weaknesses in proprietary and fine tuned LLM transferability is insufficient, and more detailed analysis and exploration of more dimensions are needed.
2. Section 6 is relevant to LLM's inquiry about linguistic competence. But is it clear to the authors whether there is multilingual data for the training data of the language model in Table 3. This will affect most of the experimental analysis in this paper.
3. This work should introduce more language types for performance analysis.
4. In Table 9 (Hard) llama3.1 70B significantly outperforms other models (e.g., 405B) in Korean, but underperforms in English instead. The authors do not seem to have made an in-depth analysis, which is regrettable.

**Questions:**

1. Could the authors provide more details on the types of factual inaccuracies and cultural misrepresentations encountered?
2. Conclusions about the multilingual generation of Section 6 need to be supported by more evidence.
3. Could the results in Table 9 give a more detailed analysis of the causes.
4. Based on the experimental findings in this paper, what specific recommendations can the authors provide for improving the performance of LLMs?

---

### Official Review · Reviewer_hBKj · 2024-11-08

**Soundness:** 3
**Presentation:** 3
**Contribution:** 2
**Rating:** 3
**Confidence:** 4

**Summary:**

This paper examines the efficacy of large language models (LLMs) as automated evaluators, commonly referred to as "LLM-as-a-Judge" models, alongside reward models (RMs) used for evaluating long-form responses. The primary focus is on their performance in diverse contexts, such as non-English languages, factual accuracy, and complex reasoning tasks. To investigate these models' capabilities and limitations, the authors introduce KUDGE, a bilingual (Korean-English) meta-evaluation dataset.

**Strengths:**

**Dataset Release**: The paper offers a thorough investigation into the capabilities and limitations of LLMs as evaluators. By introducing the KUDGE dataset, a bilingual (Korean-English) benchmark, the authors contribute an important resource for evaluating automated LLM judges, especially in non-English contexts, an area that is often underexplored. The release of the KUDGE dataset and codebase adds practical value for the broader research community (especially for Korean), enabling replication and further exploration in the field of automated model evaluation.

**Comparative study between languages**: It investigates the transferabity of new languages, which seems novel.


**High-Quality Data**: It seems it has a good data quality control and also check the ensemble of annotators.

**Weaknesses:**

**Limited interesting findings**:  There are not too many findings to take away. As a NLPer, I did not get anything to learn.

**Limited novelty for ICLR audiences**: this seems a incremental extension for MT-bench, it might better go ARR.

**on the diversity of reward models**: See more reward models on
- https://arxiv.org/abs/2410.12784 JudgeBench: A Benchmark for Evaluating LLM-based Judges
- https://arxiv.org/pdf/2403.13787 Evaluating Reward Models for Language Modeling

**Questions:**

- Could you summerize the answer for the question *WHAT THEYCAN AND CANNOT DO* as titled in the paper?

- Does the performance gap between paired compared models matters?  how is the results for a  larger gap, medium gap or a smaller gap.

- It could better show a few examples for the data.

---

### Note · Authors · 2024-11-14

**Comment:**

thanks for the sincere reviews.

**Withdrawal Confirmation:**

I have read and agree with the venue's withdrawal policy on behalf of myself and my co-authors.